# Mechanism of Action of Antitumor Au(I) N-Heterocyclic Carbene Complexes: A Computational Insight on the Targeting of TrxR Selenocysteine

**DOI:** 10.3390/ijms25052625

**Published:** 2024-02-23

**Authors:** Iogann Tolbatov, Paolo Umari, Alessandro Marrone

**Affiliations:** 1Department of Physics and Astronomy, University of Padova, Via F. Marzolo 8, 35131 Padova, Italy; paolo.umari@unipd.it; 2Dipartimento di Farmacia, Università “G d’Annunzio” di Chieti-Pescara, Via dei Vestini 31, 66100 Chieti, Italy; amarrone@unich.it

**Keywords:** anticancer metallodrugs, N-heterocyclic carbenes, gold(I) complexes, DFT calculations, proteins, thioredoxin reductase TrxR

## Abstract

The targeting of human thioredoxin reductase is widely recognized to be crucially involved in the anticancer properties of several metallodrugs, including Au(I) complexes. In this study, the mechanism of reaction between a set of five N-heterocyclic carbene Au(I) complexes and models of the active Sec residue in human thioredoxin reductase was investigated by means of density functional theory approaches. The study was specifically addressed to the kinetics and thermodynamics of the tiled process by aiming at elucidating and explaining the differential inhibitory potency in this set of analogous Au(I) bis-carbene complexes. While the calculated free energy profile showed a substantially similar reactivity, we found that the binding of these Au(I) bis-carbene at the active CysSec dyad in the TrxR enzyme could be subjected to steric and orientational restraints, underlining both the approach of the bis-carbene scaffold and the attack of the selenol group at the metal center. A new and detailed mechanistic insight to the anticancer activity of these Au(I) organometallic complexes was thus provided by consolidating the TrxR targeting paradigm.

## 1. Introduction

Metal complexes with N-heterocyclic carbene (NHC) ligands are ubiquitously employed as efficacious catalysts [1,2]. The robust metal-NHC bonds allow these complexes to gain a substantial stability in the physiological milieu [3,4] and pave their potential employment in therapy [5,6]. Inspired by the success of auranofin, the first patented gold-based drug with a plethora of anticancer, antiparasitic, antibacterial, and antiviral properties [7,8], the metallodrug community developed a gold(I)-based NHC scaffold (see, for example, Figure 1) that is characterized by the presence of a Au(I) center coordinated by two NHC ligands [9], in which the nitrogen atoms can bear the decorations of considerable size, for instance, acridine or hexyl [10]. This class of complexes demonstrates a vigorous cytotoxicity both in vitro and in vivo [11,12], which is predominantly explained on the basis of a high affinity of Au(I) toward selenols and thiols [13]. It is expected that Au(I) substitutes essential metals in places where the interaction with selenol or thiol groups is attainable, thus impeding various metabolic routes, including the redox balance and increasing the permeability of the cell membrane [11,14,15]. Other hypotheses on its possible mode of action include the direct damage to DNA, proteasome inhibition, or specific kinases modulation [16,17,18].

An example of the Au(I) complex targeting a Cys in the active site is the inhibition of the glutathione reductase (GR), which was corroborated by the crystal structure showcasing two thiols metalated by Au(I) center with nearly a linear S-Au-S coordination [19,20,21]. The most prominent cancer-related target of Au(I) is the selenocysteine (Sec) residue in the active site of the thioredoxin reductase (TrxR). Long just a hypothesis, the inhibition of TrxR by Au(I) complexes was recently proved by extended X-ray absorption fine structure (EXAFS) spectroscopy [22]. TrxR is overexpressed in the cancer cells and constitutes an excellent biotarget for the Au-based metallodrugs [23,24].

Recently, a detailed theoretical study was published that focused on the mechanism of the attack of the Au(I)-bis-NHC complex onto the Sec protein residue [25]. The model used the neutral and deprotonated capped Sec in the reaction with the phosphate buffer and without. It was shown that only the usage of deprotonated nucleophiles together with the buffer-assistance allowed for a thermodynamically favorable and kinetically feasible reaction with an estimated activation barrier lower than 19.8 kcal/mol [25]. The explanation was based on the role played by the proton transfer from the buffer to the leaving carbene moiety; thus, the significance of the environment was accentuated. Interestingly, these studies unveiled that the nucleophile attack of deprotonated Sec or Cys side chain to the Au(I) center occurs via the formation of a late transition state, thus posing steric restraints to the approach of the bis-carbene at the catalytic site of TrxR and evidencing the crucial role played by the conformational flexibility of the protein target. To test the capability of our computational model in discriminating the reactivity of a congeneric set of bis-carbene, we applied this proposed reaction pathway to a set of five cytotoxic gold(I) complexes (Figure 1) developed by the group of Ingo Ott [26,27]. The selected compounds demonstrated different inhibitory abilities, with the complex **4** found to be highly active against tumor growth, thus constituting an efficacious inhibitor of mammalian TrxR [26].

To shed light onto the difference in the TrxR inhibition activity of these complexes, we performed calculations based on the presence of the phosphate buffer. Unlike the strategies adopted to explore the chemical space around small organic active molecules, the rational design of metallodrugs requires the employment of quantum chemistry methodologies for two reasons: (i) the drug-target process occurs via ligand exchange reactions in which the electronic structure of the metal center plays a pivotal role; (ii) preactivation or assistance mechanisms could be necessary to induce the leaving of labile ligand and allow the coordination of the metal center to the target. In this view, the set of Au(I)-bis-NHC complexes is exemplificative because the inhibition of TrxR occurs via the coordination of the NHC-Au(I) moiety assisted by the protonation of the departing NHC ligand.

The pseudomolecule approach employed in the present investigation (vide infra) allowed us to model the targeting process, i.e., the binding at the side chain of a Sec residue, for each of the **1**–**5** Au(I) complexes and to analyze both thermodynamics and kinetics at a full density functional of theory level of theory. Therefore, the electronic structures of the **1**–**5** complexes, as well as those of the species intercepted along the reaction pathway, were used to rationalize the experimentally detected TrxR inhibitory trend. 

Furthermore, we developed a competing model, based on the fact that the residue Cys is neighboring to Sec, thus it is a possible assistant to the attack of the Au(I)-bis-NHC complex in place of phosphate. The CysSec dyad not only resumes the orientational restraints that the selenothiol and thiol groups are subjected to, but also constitutes a chemical model specific of the TrxR active site. In fact, our results ruled out the assistance of the thiol group and showed that the amino group of the Cys residue effectuates only a slight stabilization of the released carbene by a polar interaction with the leaving carbon. The comparison of the assistance by phosphate and Cys is discussed in light of the obtained energy profiles and of the electronic properties of the leaving carbene. A new insight on to the mechanism of TrxR targeting by bis-carbene Au(I) complexes is provided, including the possible role of NHC decoration in the modulation of the TrxR inhibition.

## 2. Results

All studied complexes (**1**–**5**) (Figure 1) were sketched into a similar geometry (Figure 1) by assuming the same core scaffold. In particular, the conformational search for complex **4** was performed to determine the optimal dihedral angles between the planar moieties, i.e., carbene-carbene and carbene-(4-F-phenyl) dihedral angles, and the rotation of the N-ethyl groups. The structures of complexes **1**–**3** and **5** were then assigned the same core scaffold of **4** by changing the substituent R. 

The reaction of complexes **1**–**5** with the selenothiolate group in the presence of the phosphate buffer was preliminarily investigated. As reported in our recent theoretical study [25], the ligand exchange reaction at an Au(I)-bis-carbene complex may favorably occur through the attack of an anionic nucleophile only if a proton is transferred from the acidic component of the buffer to the departing carbene. The presence of either a buffer or a species providing the proton assistance is necessary both in vitro and in vivo, unveiling the importance of the environment to the reaction course.

We used phosphate as a buffer since this species is present in the cytoplasm of all cells and is also one of the most employed buffers for in vitro reactions between these gold complexes and enzymes, peptides, or simple amino acids. We considered the acidic component of this buffer at neutral pH, i.e., the dihydrogen phosphate anion H_2_PO_4_^−^. The carbene substitution reaction was modelled as shown in Figure 2. The pseudomolecular approach was employed to model the reaction between each bis-carbene complex and the anionic Sec residue. Indeed, we envision that the formation of the encounter complex, i.e., reactant adduct (RA), resembles the approach of the gold(I) complex at the TrxR active site. That is, our model of reaction thermodynamics and kinetics does not incorporate any enthalpy or entropy contribution related to the ligand-enzyme interactions and/or induced fit effects. Therefore, the pseudomolecular model allows us to circumvent the errors originating in the inadequate treatment of solvation effect and entropy corrections when the infinitely separated species are considered. The use of ethylselenolate in place of the capped Sec residue was reputed more adequate as reported in [3,28].

Our calculations indicated that all complexes are highly reactive (all activation free energy values <15 kcal/mol) toward the buffer-assisted substitution of carbene by ethylselenolate (Table 1, Figure 3 left), suggesting that these complexes may target the Sec residue via a thermodynamically controlled process. 

The experimental evidence is more thermodynamic than kinetic; indeed, the IC50 values relative to the inhibition of the rat TrxR can be correlated with the amount of carbene-bound enzyme [27].

In particular, the IC50 data could be related to the constant of the enzyme-inhibitor dissociation:EI⇋E+I
and represents the molar concentration of I that binds at the 50% of the total concentration of the enzyme, *C_e_*, so that
$$\left[\mathrm{EI}\right]=\left[E\right]=\frac{C_{e}}{2}\;\mathrm{with}\;\left[I\right]=\mathrm{IC}50$$

Thus, by inserting the tiled relations in the expression of the dissociation constant *K_d_*:$$K_{d}=\frac{\left[E\right][I]}{\left[\mathrm{EI}\right]}=\frac{\frac{C_{e}}{2}\;\mathrm{IC}50}{\frac{C_{e}}{2}}=\mathrm{IC}50$$

Consequently, the standard free energy for the above process depends on the dissociation constant and it can be written as
$$∆G^{0}=-RTln{(K}_{d})=-RTln(\mathrm{IC}50)$$

Thus, we took the −Δ*G*^0^ values derived from the IC50 data, which reflect the association process, and can be better compared with the calculated RA → PA free energies for the **1**–**5** complexes. 

As shown, the calculated RA → PA free energies were all positive whereas the enzyme-inhibitor association free energies derived from the IC50 data were all negative. Such an apparent discrepancy can be explained by noticing that the RA → PA process is not the whole, but only one of the stages of, the enzyme-inhibitor association process. For example, the conformational adaptation of a macromolecule as the TrxR enzyme to the bis-carbene binding may yield a quite significant free energy contribution to the association process. Moreover, when the real process is concerned, a non-covalent enzyme-inhibitor adduct may form in which the R substituents of the carbene ligands can lead to further stabilization by accessing specific binding pockets. In the first approximation, we assumed that all **1**–**5** ligands induced a similar conformational and steric adaptation in the targeted TrxR, so that the calculated RA → PA and the IC50-derived free energies have almost the same variance. 

In this framework, we more accurately compared the relative free energy values obtained by subtracting each entry in Table 1 with the value for the fluorinate complex **4** (see data in parentheses). As shown, we found an overall qualitative agreement between calculated and experimental values, with the only discrepancy detected for complex **5**.

The fact that the Br derivative **5** has the most favorable thermodynamics in the reduced model while the IC50 derived data indicated this complex is less effective than **4** by 1.83 kcal/mol in the binding at TrxR, could be related to the higher steric hindrance of Br compared to F. Indeed, the approach of the bis-carbene to the Sec residue could be related to the access of the R substituent in a hydrophobic pocket, i.e., presumably the F-Ph group of **4** bears an optimal size whereas the larger Br-phenyl of **5** requires some more induced-fit costs. Thus, the bioactivity of these gold(I) complexes seems to be majorly affected by the steric hindrance of the substituents on the NHC ligand and by the induced conformational adaptation of the presumed target, i.e., TrxR. To better elucidate this aspect, we computationally investigated the departing group attitude of the NHC ligands (Table 2). 

As shown, both the frontier orbitals’ energy and the carbene atom charge were found in quite narrow ranges, evidencing that the five NHC ligands bear overall similar electronic features. Thus, we expect that these scaffolds are characterized by almost similar leaving group features. 

## 3. Discussion

Our calculations showed how buffer assistance is very effective in promoting the release of the NHC ligand and the formation of the Au-Se bond. On the other hand, the approach and incorporation of the buffer moiety in the encounter complex poses further structural restraints to the viability of this mechanistic hypothesis. Thus, we questioned whether the protein environment itself may provide for some assistance and, thus, facilitate the approach and the reaction of Au(I) bis-carbene complexes with the active Sec of TrxR. In the present investigation, we focused on the possible mechanistic role played by both the upstream cysteine and the backbone groups vicinal to the targeted selenothiolate group. In particular, we assumed that the CysSec dyad presented the protonated thiol and deprotonated selenol groups at physiological pH, with a weakly acidic thiol group of the Cys proximal to the aurophilic Sec side chain and prompted to assist the carbene release.

The reaction between **4** and the CysSec dyad was then investigated at the DFT level of theory. The conformational space of CysSec was preliminary explored to gain a representative structure of the peptide in aqueous solution, based on its high bulk exposure [29]. 

The TS structure of the tiled process was constructed by approaching the CysSec to the bis-carbene scaffold and by assuming the assistance of the Cys thiol to the carbene release, i.e., by orienting the S–H bond toward the leaving carbon atoms. The calculated free energy profile of the reaction between **4** and the CysSec dyad showed that the activation barrier of 16.2 kcal/mol is higher than the barrier for either the phosphate-assisted or the non-assisted mechanism (Figure 3). The comparison of the corresponding TS structures evidenced how the conformation of the CysSec peptide exerts a significant steric restraint to the approach and reaction of the bis-carbene moiety (Cartesian coordinates of TS structures can be found in Appendix A). Despite the lengths of the forming Au-Se bond are similar in the compared mechanisms and are consistent with late TS structures, the Se-Au-C(leaving) angle in the reaction with CysSec is much higher, probably because of the steric influence exerted by the peptide structure. Therefore, if an analogous spatial orientation of the bis-carbene scaffold is assumed, the direction of the selenolate moiety attack in the reaction of **4** with the CysSec dyad is completely different compared to either phosphate-assisted and non-assisted mechanisms (Figure 4), thus testifying the crucial role played by the TrxR structure. The calculated transition state structures for the reaction of **4** also evidenced that, independent of the mechanistic hypothesis, the substituents on the positions four or five of the NHC ligands are projected far from the reaction center, thus only marginally influencing the carbene release.

Above all, our computational results highlighted two aspects of the TrxR inhibition by Au(I) bis-carbene complexes. On the one hand, steric interactions are prominent in shaping the spatial approach and the reaction between Au(I) bis-carbene complexes and the active Sec residue on the TrxR, probably softening the high reactivity disclosed by the selenolate moiety. On the other hand, the substituents of the NHC ligands may be involved in the stabilization of the encounter complex between the bis-carbene complex and the TrxR enzyme by accessing available hydrophobic pockets.

## 4. Materials and Methods

All the calculations were conducted by means of the Gaussian 16 quantum chemistry software [30]. DFT permits the accurate description of transition metal-based complexes with biomolecules [31,32,33], also containing coinage metals [34,35,36] and, notably, gold(I) [37,38,39]. The optimization in water was conducted with the hybrid range-corrected functional ωB97X-D [40] for all structures. Indeed, this functional is recognized to produce correct geometrical structures and to accurately compute the electronic and solvation energies [41,42,43]. The optimization was performed with the basis def2SVP, while the computation of electronic and solvation energies was completed with a more flexible triple-zeta quality basis set def2TZVP [44,45]. The stationary character of the minima and saddle points was substantiated by the frequency computations which also yielded the zero-point energy (ZPE) as well as vibrational corrections to the thermodynamic properties. The reactant-adduct (RA) and product-adduct (PA) minima were found via the usage of intrinsic reaction coordinate (IRC) methodology and afterwards optimized. The ZPE, thermal, and entropy corrections as well as the Gibbs free energy values were computed by means of the harmonic approximation. The solvation was described by the IEFPCM technique [46,47], a method producing accurate values for the free energies, compared to the other continuum models, both for neutral and charged molecules [48,49]. 

The conformational space of the capped CysSec dyad, i.e., N-acetyl-Cys^1^Sec^2^-C=O(NHCH_3_), was explored by using the crest software (version 2.12) [50]. The CysSec peptide was assumed in its mono anionic form, with neutral Cys thiol and deprotonated Sec side chain, and surrounded by 50 water molecules. 

The water cluster and the conformational exploration were performed by using the qcg module of crest, at the tight-binding DFT level of theory implemented in xTB [51]. An ellipsoid wall potential (diameters assigned by crest: 18.79603388222205, 17.84202764346258, and 15.66339037421275 Å) and the generalized Born implicit solvation method were employed to exert the steric and polarization effects of the aqueous bulk, respectively.

## 5. Conclusions

The gold(I)-based metallodrugs exhibit a promising antitumor activity. There is a general consensus in the medicinal chemistry community that they target the Sec of mammalian TrxR. However, the details of the attack, i.e., the mechanistic machinery yielding the formation of the Au-Se bond, differ for various gold complexes. The attack may be facilitated either by the assistance of a buffer moiety or via the electronic and steric effects of the metal scaffold itself. Thus, the structure-based rationalization of the TrxR inhibition by bis-NHC-Au(I) complexes required the employment of quantum chemistry methodologies, which can model the ligand exchange process and the concomitant buffer assistance.

For this purpose, the pseudomolecule approach allowed us to adequately describe the thermodynamics and the kinetics of the tiled processes and highlight the structure-reactivity relationships within the analyzed **1**–**5** complexes. In previous studies, five possible mechanisms were outlined, according to which the gold(I)-bis-carbene attack on Sec of TrxR takes place. However, the phosphate-assisted mechanism, which was found to be the most probable, does not allow to exhaustively describe the IC50 values’ trend in bis-carbene complexes with differing phenyl- or halide-containing ligands. In order to obtain a consistent result, we developed a model based on the CysSec dyad, which reproduces the CysSec dipeptide found in the active site of TrxR. The computations have shown that although this competing model was investigated to assess the possible Cys assistance to the release of an NHC ligand, the stabilization offered by the Cys thiol is only marginal. Therefore, despite less reactive compared to the bare ethylselenolate ligand, calculations showed how the reaction between the CysSec peptide and **1** is characterized by an activation free energy of only 16 kcal/mol. The steric and orientational restraints, occurring when the reaction of CysSec diad is concerned, are probably crucial in the stage of the formation of the encounter complex by leading the substituents on the NHC scaffolds to access specific interaction pockets. Above all, the results of the present study shed new light on the mechanism of TrxR inhibition by Au(I) bis-carbene by also paving the way to more specific addresses of investigation. Indeed, while the substitution of the NHC ligand by the TrxR selenolate group was found to be kinetically facile, our computational model evidenced how the steric effects may play a pivotal role by controlling the orientation of the bis-carbene complex toward the targeted Sec residue. Thus, we envision that further investigations should be devoted to clarifying the steric aspect. Based on the results of the present study, we repute that the investigation of asymmetric bis-NHC-Au(I) complexes, for instance, the R-NHC-Au(I)-X complexes bearing labile ligand X, would better yield the structure-activity relationships affecting the labile and the carrier metal ligands. 

## Figures and Tables

**Figure 1 ijms-25-02625-f001:**
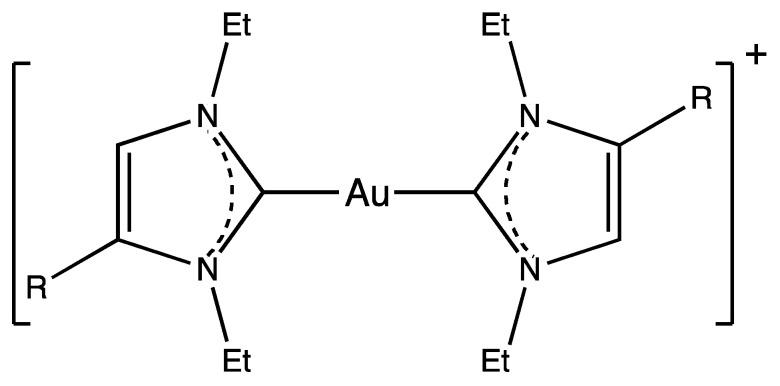
The studied complexes **1** (R=H), **2** (R=Br), **3** (R=Ph), **4** (R=F-Ph), and **5** (R=Br-Ph).

**Figure 2 ijms-25-02625-f002:**
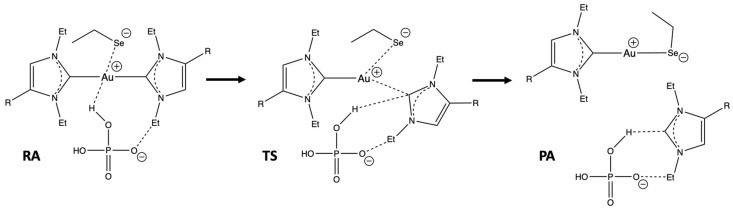
Pseudomolecular model of the carbene substitution reaction in complexes **1**–**5** by ethylselenolate assisted by dihydrogen phosphate. RA, TS, and PA stand for reactant-adduct, transition state, and product-adduct, respectively.

**Figure 3 ijms-25-02625-f003:**
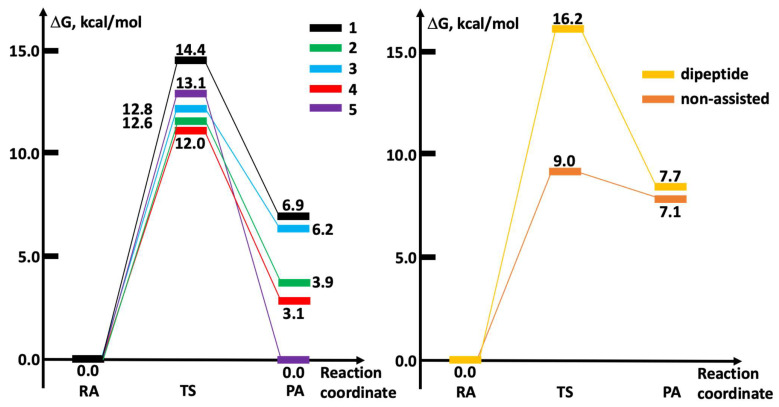
Calculated Gibbs free energy profiles for the reaction of **1**–**5** complexes with ethylselenolate via phosphate assistance (**left**), and for the reaction of **1** with the CysSec dipeptide or with (non-assisted) ethylselenolate (**right**). All values in kcal/mol.

**Figure 4 ijms-25-02625-f004:**
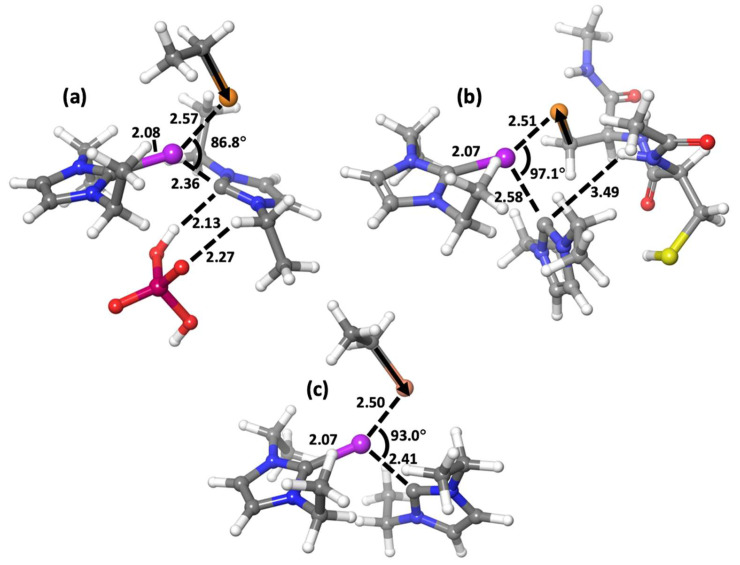
Rendition of the calculated transition state structure for the reaction of 1 with ethylselenolate via no assistance (**c**), phosphate buffer assistance (**a**), and for the reaction with the CysSec peptide (**b**). The forming/breaking coordination bonds and the assisting interactions are reported as dashed lines. The orientation of the C-Se bond of each nucleophile (corresponding to the orientation of the attack) is also depicted (black arrow). Bond distances in angstrom. Color scheme: Au (plum), Se (orange), S (yellow), P (burgundy), O (red), N (blue), C (grey), H (white).

**Table 1 ijms-25-02625-t001:** Calculated free energies for the RA → TS and RA → PA processes and estimated free energies for the enzyme-inhibitor association reaction derived from IC50 data. Relative free energy values are reported in parentheses. All values in kcal/mol.

Complexes	RA → TS	RA → PA	−Δ*G*^0^ (from IC50)
**1**	14.4 (+2.4)	6.9 (+3.8)	−5.31 (+3.01)
**2**	12.6 (+0.6)	3.9 (+0.8)	−5.36 (+2.96)
**3**	12.8 (+0.8)	6.2 (+3.1)	−6.14 (+2.18)
**4**	12.0 (0.0)	3.1 (0.0)	−8.32 (0.0)
**5**	13.1 (+1.1)	0.0 (−3.1)	−6.49 (+1.83)

**Table 2 ijms-25-02625-t002:** Energy of the highest occupied (HOMO) and lowest unoccupied (LUMO) molecular orbitals (in hartree), the HOMO-LUMO gap (in eV), and the Mulliken charge (in electron units) on the leaving carbon atom are reported.

Complex	HOMO, a.u.	LUMO, a.u.	HOMO-LUMO Gap, eV	Charge on Carbene Carbon
**1**	−0.313	0.094	11.1	−0.349
**2**	−0.309	0.071	10.3	−0.327
**3**	−0.301	0.036	9.2	−0.350
**4**	−0.301	0.037	9.2	−0.349
**5**	−0.301	0.026	8.9	−0.346

## Data Availability

Data are contained within the article and Appendix A.

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
