# Peer review of "Mechanism of Action of Antitumor Au(I) N-Heterocyclic Carbene Complexes: A Computational Insight on the Targeting of TrxR Selenocysteine"

_ijms, 2024, doi:10.3390/ijms25052625_

Round 1

Reviewer 1 Report

Comments and Suggestions for Authors

The study's objective was to elucidate the kinetics and thermodynamics of the reaction between these complexes and the active Sec residue in TrxR, providing insights into the differential inhibitory potency among a set of analogous Au(I) bis-carbene complexes. The study provides a comprehensive understanding of the antitumor activity of gold(I)-based metallodrugs, confirming their targeting of the Sec of mammalian TrxR. It offers new insights into the mechanistic machinery leading to the formation of the Au-Se bond, suggesting that the reaction may be facilitated by buffer moieties or through electronic and steric effects of the metal scaffold. After addressing the following minor issues, this work can be accepted.

1) The author should explain more about the estimation of IC50 = Kd.

2) Due to the existence of noncovalent interactions, the low-frequency mode should be considered to calculate the Gibbs free energy.

Reviewer 2 Report

Comments and Suggestions for Authors

Iogann et al. submitted the manuscript entitled: Mechanism of action of antitumor Au (I) N-heterocyclic carbene complexes: a computational insight on the targeting of TrxR selenocysteine, in which they dug into the MOA of Au-NHC complex with the aid of computational chemistry. While I think this work might be of interest to the potential readers of IJMS, I have some concerns as follows regarding the project design of this manuscript.

1. The authors should not directly consider cell apoptosis as cell engagement of TrxR, as other drugabilities like cell permeability, or in vitro stability, will also have influences on compound efficacy.

2. Figure 1: since substitution of R contribute to the inhibition fluctuation. The authors may try to dig deeper into the relation between TrxR binding affinity and electronic potential of functional groups at this position. Based on the existing findings, the authors can try to design and predict some new compounds bearing this skeleton.

3. For the DFT calculation, the authors can consider include calculations of NHC-Au-Cl as a comparison.

Round 2

Reviewer 2 Report

Comments and Suggestions for Authors

The authors have well addressed on all the issues.